# Pediatric Extracranial Germ Cell Tumors: Review of Clinics and Perspectives in Application of Autologous Stem Cell Transplantation

**DOI:** 10.3390/cancers15071998

**Published:** 2023-03-27

**Authors:** Chong-Zhi Lew, Hsi-Che Liu, Jen-Yin Hou, Ting-Huan Huang, Ting-Chi Yeh

**Affiliations:** 1Division of Pediatric Hematology-Oncology, Department of Pediatrics, Mackay Children’s Hospital, Mackay Medical College, Taipei 104, Taiwan; 2Division of Pediatric Hematology-Oncology, Department of Pediatrics, Hsinchu Mackay Memorial Hospital, Hsinchu 300, Taiwan

**Keywords:** pediatric, extracranial, germ cell tumor, high-dose chemotherapy, autologous stem cell transplantation

## Abstract

**Simple Summary:**

Poor-risk subgroups of pediatric germ cell tumors may have poor responses to conventional first-line chemotherapy. This review examines the current clinical perspectives of high-dose chemotherapy combined with autologous stem cell transplantation in children with relapse or refractory extracranial germ cell tumors and assesses the feasibility of applying autologous stem cell transplantation in high-risk patients with germ cell tumors.

**Abstract:**

Pediatric extracranial germ cell tumors (GCTs) are rare, accounting for approximately 3.5% of childhood cancers. Since the introduction of platinum-based chemotherapy, the survival rate of patients has improved to more than 80%. However, poor-risk subtypes of pediatric extracranial GCTs do not respond well to chemotherapy, leading to refractory or relapsed (R/R) diseases. For example, long-term survival rates of mediastinal GCTs or choriocarcinoma are less than 50%. According to reports in recent years for adult patients with R/R GCTs, the use of high-dose chemotherapy (HDCT) combined with autologous stem cell transplantation (ASCT) has clinical advantages; however, HDCT combined with ASCT has rarely been reported in pediatric GCTs. The R/R and poor-risk groups of pediatric GCTs could benefit from HDCT and ASCT.

## 1. Introduction

Pediatric germ cell tumors (GCTs) are a rare type of childhood cancer, accounting for only 3.5% of all pediatric neoplasms [1]. The age distribution of Pediatric GCTs has a bimodal age distribution pattern: that is, the incidence peaks before 4 years of age and between 15 and 19 [2]. The treatment strategy for pediatric GCTs is complete resection followed by adjuvant chemotherapy. If the tumor cannot be completely resected at the time of diagnosis, a biopsy followed by neoadjuvant chemotherapy is usually performed. The purpose of the above is to completely resect the residual tumor. Pediatric extracranial GCTs generally have a good prognosis, with 10-year overall survival (OS) and event free survival (EFS) rates of 95% and 88%, respectively [3]. However, some subtypes of extracranial GCTs are resistant to platinum-based chemotherapy, leading to poor prognosis [4]. For those patients classified as high risk, a cure rate of less than 70% was reported [4]. High-dose chemotherapy (HDCT) combined with autologous stem cell transplantation (ASCT) has been proven to have potential therapeutic benefits in adult relapse or refractory (R/R) GCTs patients [5]. Currently, there is no consensus on the indications of HDCT combined with ASCT in pediatric GCTs or the conditioning regimens before ASCT. The primary objective of this review is to understand an optimal conditioning regimen, treatment toxicities, and prognosis of HDCT combined with ASCT in pediatric GCTs patients. The second aim is to define the indications for HDCT combined with ASCT in pediatric GCTs and determine whether this combination offers a potentially advantageous treatment option. Our systematic review followed the Preferred Reporting Items for Systematic Review and Meta-Analysis (PRISMA) guidelines, and the registration number is INPLASY202310081.

## 2. Pediatric Extracranial Germ Cell Tumor

### 2.1. Prevalence and Embryogenesis

The prevalence of pediatric GCTs is correlated with race, accounting for 3.5% of all childhood cancers in Western countries and 5–11% in Asian countries [1,6]. Primary extracranial GCTs included gonadal and extra-gonadal disease [7]. Gonadal and extragonadal tumors equally occurred before 4 years of age, whereas the majority of the tumors diagnosed after 10 years old were gonadal disease [2]. Among the two main subtypes of pediatric GCTs, testicular GCT accounts for between 1% and 2% of all childhood cancers, with the highest incidence in European countries, suggesting a genetic predisposing factor [8,9,10,11]. Ovarian GCTs primarily affected adolescents and young women [12]. Incidences increase from 8 to 9 years of age. East Asia reported the highest incidence [13].

Primitive germ cells migrate during embryogenesis from their origin in the endoderm in the yolk sac through the midline to the urogenital ridge and the gonads. Abnormal migration results in distribution of germ cells inside and outside of the gonads [14,15]. Ideally, germ cells die once they are misplaced; however, if the misplaced germ cells persist without apoptosis due to various factors, such as genetic alternations in primordial germ cells or abnormal changes in the microenvironment where the cells are located, it may lead to the development of further GCTs [16]. Figure 1 illustrated that the occurrence of extragonadal GCTs and their midline propensity were due to this disruption of the migration process. The majority of pediatric primary extracranial GTCs occur in the extragonadal site, which is limited to the midline structures, such as the sacrococcygeal, retroperitoneal, mediastinal, cervical, and intracranial regions [17]. GCTs also can be further classified into type I and type II according to the cell’s age at presentation and histology features [17]. Generally, early primordial germ cells are those which migrated to the genital ridge and underwent maturation into a gonadal germ cell, then developed into future gonads [17]. Primordial germ cells that are misplaced outside the gonads should undergo apoptosis. However, the escaped apoptosis primordial germ cells could give rise to various malignant GCTs; they are usually located at the migration pathway, midline of the body [17]. Type I GCTs are often found in children younger than age four with benign teratoma or yolk sac tumors (YST) [17]. In addition, Type II GCTs often arise around the age of puberty. Type II GCTs have been associated with a precursor lesion known as germ cell neoplasia in situ (GCNIS), a testis-specific lesion [17]. The molecular mechanisms of GCNIS leading to invasive cancer transformation are still under investigation.

Pediatric GCTs are divided into two categories based on histopathological classification, including germinomas and non-germinomatous GCTs. The precursor cells of germinomas remain undifferentiated; they resembling primitive germ cells and are known as seminomas in males and dysgerminomas in females. Non-germinomatous GCTs consist of embryonic tissue, such as teratoma and embryonal carcinoma (EC), and extraembryonic tissue, such as YST and choriocarcinoma (CC) [7]. The most common gonadal GCTs are YST in male and teratoma in female young children [18]. Mixed GCTs have more than one type of histology tissue; these are common and make up 32–60% of all GCTs [19]. The most common admixtures were EC and teratoma [19].

Early PGC migration from yolk sac to the genital ridge and along the midline of the body. The process involved methylation erasure. When the early PGC escape apoptosis which may have associated gene regulations or epigenetic microenvironment influence, they could transform into GCT I, which was usually teratoma or YST. Then, the early PGC underwent maturation and transformed into gonadal PGC in the gonads. The gonadal PGCs could undergo malignant transformation and become the precursor of germ cell tumors. If this pathological differentiation process occurs in testis, it is called GCNIS; in the ovary, it is called gonadoblastoma. GCNIS or gonadoblastomas can give rise to various histology types of GCT II when associated with gene mutations or epigenetic gene regulations. 

### 2.2. Genetics and Molecular Biology

Pediatric GCTs are hypothesized to develop gradually during the fetal period, although the exact cause of this disease remains largely unknown. Previously, reports confirmed a strong genetic susceptibility to the disease [18]. For example, the association between sex chromosome abnormalities and the development of pediatric GCTs is well documented [16]. Patients with Klinefelter syndrome (KS), chromosome 47, and XXY are at increased risk of developing extragonadal GCTs, especially in the mediastinum and peritoneum [16]. Reports from the Cancer Oncology Group (COG) demonstrated one-third of patients with mediastinal GCTs had KS [16,20]. Therefore, determining if a child with mediastinal GCTs has the KS karyotype can help to understand and manage other diseases in the future. 

Adult testicular GCTs were found to have gained chromosomes 12p, usually as an isochromosome 12p formation. However, this is less frequently observed in children [18,21]. Other genetic alterations that have been shown to be associated with GCTs include; the germinomas exhibit global hypomethylation and high expression of POU5F1 or NANOG; isochromosome 12p; or KIT mutation; TNFRSF8 is characteristic of EC [22]. YSTs often overexpress endodermal genes, such as FOXA2, HNF4A, ERBB4, and GATA6 [22]. TP53 mutation can occur in mediastinal GCTs in older children [22]. As the causative gene alternations in GCTs appear to be heterogeneous, further studies are needed to confirm these molecular targets.

### 2.3. Histology Classification and Staging System

#### 2.3.1. Seminoma/Dysgerminoma

Dysgerminomas in the ovary and seminomas in the testis have similar histomorphologies [23]. Seminomas frequently occur in young men between the ages of 15 and 35 and are the most common testicular GCTs in this age group [24]. Dysgerminomas account for only about 2% of ovarian neoplasms [25]; most occur before the first two decades of life [26]. Histopathologically, these tumors appear as a nested arrangement separated by bands of fibrous tissue that contain variable numbers of lymphocytes. When composed of syncytiotrophoblasts, these tumors will secrete β-HCG [23].

#### 2.3.2. Teratoma/Immature Teratoma

Teratomas are divided into mature and immature teratomas according to histopathology. Arising from totipotent germ cells, mature teratomas consist of well-differentiated tissue from all three germ layers. Immature teratomas usually contain immature neuroepithelial tissue in addition to tissue from all three germ layers [27]. Complete resection is crucial for the treatment of mature and immature teratoma [28]. Those with incomplete surgical resection, especially of sacrococcygeal teratomas, often relapse despite effective cisplatin-based chemotherapy [28]. Mature and immature teratomas most commonly occur in the ovaries of girls around puberty [27]. Extragonadal teratomas more commonly occur in young children. For example, sacrococcygeal teratomas, the most common tumors in newborns, occurred in 1 in 27,000 births, with female predominance at a ratio of 3–4:1 [29].

#### 2.3.3. Yolk Sac Tumor (YST)

Histopathologically, YST is characterized by friable and myxoid tissue with varying degrees of hemorrhage and necrosis [30]. Schiller–Duval bodies, which are the central vessel tightly embedded with tumor cells, are an important feature [30]. YST is the most common purely malignant GCT in children, accounting for 55% of pediatric GCTs [4]. The peak incidence of YSTs is within the first three years of life and occurs most frequently in the testes [31]. Compared with prepubertal YST, postpubertal YST usually occurs as a mixed GCT with a higher degree of malignancy that is often more resistant to chemotherapy [32,33].

#### 2.3.4. Embryonal Carcinoma (EC)

The histology of EC is characterized by large cells with large, overlapping nuclei and very large nucleoli [23]. EC cells are negative for alpha-fetoprotein (AFP), positive for OCT-4, and positive for CD30 using immunohistochemical staining [23]. EC are non-seminoma stem cells that can differentiate into YST, CC, or teratomas [34]. EC is relatively common in testicular GCTs but rare among ovarian GCTs [34].

#### 2.3.5. Choriocarcinoma (CC)

CC consists of two types of cells, cytotrophoblasts and syncytiotrophoblasts. The positive staining of β-HCG in the syncytiotrophoblast tissue of CC is related to the high serum **β**-HCG level at the time of diagnosis. CC is fragile and bleeds easily. Most patients have already developed metastases at diagnosis, and the prognosis is poor among pediatric GCTs [35,36,37].

#### 2.3.6. Mixed Germ Cell Tumors

Mixed GCTs are the second common histology type of malignant GCTs, accounting for around 40% [4,35]. Mixed malignant GCTs consist of more than two histology types of malignant germ cells and are more frequent with increasing age [23].

#### 2.3.7. Staging System

Malignant GCTs in children are staged according to the Children’s Oncology Group (COG) system. This systematic staging is based primarily on imaging findings, postoperative pathological findings, lymph node involvement, and the presence of metastases [18]. Generally, stage I refers to completely resected tumors; stage II, microscopic residual disease or a persistent raised tumor marker after resection; stage III, gross residual disease or nodal involvement; and stage IV for distant metastases [18]. However, there is a debate over inconsistencies in the staging systems. For example, an ovarian tumor with positive peritoneal cytology would be classified as stage IC by the International Federation of Gynecology and Obstetrics (FIGO) staging system, but the COG staging system would classify it as stage III. Despite inconsistencies between the different staging systems, treatment of pediatric malignant GCTs is primarily stratified by the Malignant Germ Cell Tumors International Collaboration (MaGIC) risk group based on age at diagnosis, tumor site, and COG staging [4,18]. Given the rarity of pediatric GCTs, some subgroups of patients with poor prognosis may not be clearly distinguished by this stratification system, such as those with extragonadal GCTs, especially in the primary mediastinal region. Despite multi-modality treatment with chemotherapy and surgical treatment, the 5-year EFS and OS were 40% and 54%, respectively [35]. In addition, the treatment outcome of mediastinal choriocarcinoma was extremely poor. None of the nine patients survived more than five years [35]. 

Although previous studies have found that having a high level of tumor markers before surgery is a poor prognosis factor [38,39], the MaGIC study group found that it was not significant as a factor of poor prognosis [4]. On the contrary, the decline rate of tumor markers determines whether the disease has a good response to treatment, and the results can be used to guide the treatment of patients with poor-risk GCTs [40,41].

### 2.4. Risk Classification

In addition to the aforementioned staging system, Frazier et al. defined risk classifications of pediatric GCTs according to age at diagnosis, location of tumors, stage, and survival rates [4]. In general, children with low risk or intermediate risk have a good prognosis, however, the disease-free survival (DFS) rate of children with poor-risk is less than 70% [4]. For example, the expected DFS for patients over 11 years of age with stage IV ovarian disease, stages II–III extragonadal disease, and stage IV extragonadal disease are 67%, 65%, and 40%, respectively [4]. In addition to the above patients, the 5-year OS of pediatric mediastinal GCTs is only 54%, especially in patients who have not undergone surgical resection; then, the OS is only 7% [35] (Table 1). Pediatric CC or patients without a good response to treatment, such as when tumor markers decline as expected, have a poor prognosis. Fizazi et al. found that a favorable tumor marker decline was associated with significantly better PFS and OS [40]. The favorable tumor marker decline was defined as tumor marker normalization within 9 weeks for AFP and 6 weeks for **β**-HCG, which correspond to three cycles and two cycles of chemotherapy treatment, respectively [40]. The 4-year PFS and OS were 38% and 58%, respectively, in patients who did not have a satisfactory decline of the tumor marker after starting treatment [40]. In the GETUG 13 trial, early assessment of the tumor marker was performed from days 18 to 21 to predict its decline model, and the correlation between the prognosis of chemotherapy dose-intensive treatment or standard dose. The decline of tumor markers was also analyzed. In this study, it was confirmed that poor-risk patients who continued to receive standard doses of BEP (bleomycin, etoposide, cisplatin) had poor decline in tumor markers, and the 3-year PFS and 3-year OS were 48% and 65%, respectively [41]. Therefore, the above subtypes of patients should also be classified as poor risk. As the limitations of conventional chemotherapy for the treatment of children with poor-risk, HDCT combined with ASCT, radiotherapy, or target therapy may be considered for the treatment of these types of patients.

### 2.5. Treatment Overview

#### 2.5.1. Surgical Treatment

Surgical resection is the mainstay of treatment for patients with teratomas, as these tumors do not necessarily respond well to chemotherapy. The German Pediatric MAKEI Group reported that the prognosis of patients with complete resection of mediastinal teratomas or only microscopic residuals was excellent [43]. For pediatric GCTs, if the extent of surgical resection is expected to be too large and causes subsequent serious complications, delayed resection treatment strategy after neoadjuvant chemotherapy can improve the complete resection rate. Most of the pediatric GCTs respond well to chemotherapy; therefore, adding chemotherapy will improve OS [43,44]. The current treatment strategy for pediatric GCTs is that patients with stage I or II GCTs undergo tumor resection at the time of diagnosis; patients with stage III and IV undergo surgical biopsy at the time of diagnosis, followed by chemotherapy or radiation therapy. After tumor shrinkage with treatment, the second interval debulking surgery (IDS) is performed. Whether chemotherapy is given after the second IDS should be decided according to the pathological results at the time of surgery.

#### 2.5.2. Conventional Chemotherapy and Toxicity

Chemotherapy is indispensable in the treatment of pediatric GCTs because most patients respond well to chemotherapy. In 1977, Einhorn et al. reported that cisplatin-based chemotherapy, cisplatin, vinblastine, and bleomycin (PVB protocol), was successful in the treatment of adult testicular GCTs [45]. Platinum-based chemotherapy was introduced in the treatment of pediatric GCTS in subsequent studies and successfully demonstrated improvement in survival rates [18]. Since chemotherapy significantly improves the survival rate of pediatric GCTs, research in recent years has focused on reducing the toxicity of chemotherapeutic agents. Several representative studies are as follows: Williams, et al. reported that the use of vincristine was replaced by etoposide and demonstrated that etoposide treatment has an excellent efficacy for the treatment of pediatric GCTs and that vincristine-related neurotoxicity could be reduced [46]. For patients who received the PVB protocol treatment, weekly bleomycin treatment resulted in up to 10% pulmonary fibrosis and other pulmonary toxicities [47,48]. In the Indiana University study, 22 of 166 (13%) patients had mediastinal non-seminomatous GCTs; they underwent thoracic surgery and treatment with BEP chemotherapy, then developed acute respiratory distress syndrome and experienced prolonged ventilator use [49]. It was believed there was a correlation with the weekly bleomycin use; however, the reduced dose of bleomycin, once per cycle vs. once per week, was not conducted as a head-to-head study to compare the result in the pediatric population with GCTs. Bokemeyer et al. reported reducing the weekly bleomycin dose to once per cycle, which maintained excellent therapeutic effects for GCTs. Approximately 80% of patients with metastatic GCTs were successfully treated [50]. 

The use of cisplatin in infants and children has been shown to be associated with subsequent nephrotoxicity and ototoxicity [51,52]. Patients may develop progressive high-frequency hearing loss after years of exposure to cisplatin [53]. In addition, patients may also experience paresthesia and Raynaud’s phenomenon. Therefore, the Children’s Cancer and Leukemia Group (CCLG) recommended using carboplatin instead of cisplatin as the first-line treatment of pediatric GCTs, i.e., JEb regimen (carboplatin, etoposide, and bleomycin). This combination displayed non-inferiority, and there was tolerable toxicity of carboplatin in the CCLG study. This was substantiated by subsequent reports by other study groups [18,54].

Mann et al. have proven that the JEb protocol is an effective treatment for malignant extracranial GCTs in children and has tolerable toxicity. Therefore, the JEb protocol has been widely used to treat different types of malignant GCTs [3,55]. Children undergoing carboplatin-based chemotherapy mostly receive 600 mg/m^2^ every three weeks, corresponding to a median area under curve (AUC) of 7–9 mg/mL/min. A higher carboplatin dose resulted in a JEb regimen that had comparable results to the cisplatin-based regimen. In addition, there have been no reports of sensorineural hearing loss in patients using the JEb regimen [55]. If the patient’s clinical situation necessitates avoiding possible pulmonary toxicity from bleomycin, treatment options such as VIP/PEI (cisplatin, etoposide, and ifosfamide) should be considered, based on the patient’s non-inferiority results [56]. For recurrent GCTs, salvage second-line chemotherapy is typically TIP (paclitaxel, ifosfamide, cisplatin). Regarding the toxicity of cisplatin in children, the Children’s Oncology Group (COG) used the TIC regimen (paclitaxel 135 mg/m^2^/day on day 1, ifosfamide 1800 mg/m^2^/dose on days 1–5, carboplatin AUC 6.5 mg/m^2^/day on day 1) as second-line chemotherapy, with a reported response rate of 44%, suggesting that TIC may be an alternative treatment option to TIP [57].

#### 2.5.3. Radiotherapy

Several pathological subtypes of pediatric GCTs are sensitive to radiotherapy; for example, radiotherapy for mediastinal seminoma which has an excellent survival rate [58,59]. However, radiation therapy is currently not routinely used as a first-line treatment for pediatric GCTs. This is mainly because radiation therapy can lead to long-term sequelae, including cardiorespiratory dysfunction, pulmonary fibrosis, or secondary malignancies. Wang et al. demonstrated improved 5-year OS and progression-free survival in adult malignant GCTs treated with chemotherapy and radiotherapy [60]. However, the risks and benefits of comprehensive treatment of pediatric GCTs with radiation therapy are currently unclear. Newer radiotherapy modalities, such as proton therapy, have demonstrated benefits in the treatment of pediatric intracranial GCTs and may be considered for local disease control of residual tumor after surgical resection. Whether radiation therapy has a therapeutic benefit for poor-risk pediatric GCTs requires future study [35,61].

#### 2.5.4. Targeted-Therapy and Novel Therapy

Although pediatric GCTs can be successfully treated with surgery combined with chemotherapy, there are still a small number of patients with refractory diseases. In previous reports, the patients with refractory GCTs may have responses or partial responses to targeted therapy such as tyrosine kinase inhibitors (TKIs), anti-vascular endothelial growth factor (VEGF), or anti-programmed death receptor 1 (PD-1) [62]. However, currently, no targeted therapy has been shown to be effective in the treatment of pediatric GCTs alone. Therefore, the efficacy of targeted therapy is under investigation [63]. Overexpression of vascular endothelium growth factor (VEGF) of the GCTs were noted. Previous studies showed that overexpression may have been associated with the tumor’s aggressive behavior, the propensity of tumor metastasize [63]. Nieto et al. demonstrated bevacizumab, combined with chemotherapy in treating the platinum-resistant and refractory diseases in adults, had shown encouraging results but needed further confirmation for this treatment strategy. The associated toxicity should be taken into consideration [64]. Frankhauser et al. found that the expression of PD-1 and PDL-1 were high in GCTs which may bring insight to immune checkpoint inhibitor treatment [65]. A previous report suggested that a disseminated platinum refractory non-seminomatous GCTs showed response with tumor marker decline after initiating off-label nivolumab; however the patient had new brain metastasis 5 months later [66]. A study from MD Anderson phase II cohort trial which analyzed 12 adult patients with refractory malignant GCTs diseases showed that pembrolizumab to have only limited antitumor activity with median PFS at 2.4 months [67]. 

Previous report found that cisplatin resistance disease may have a correlation with DNA hypomethylation [68]. A study conducted by Indiana University which included 14 adult patients with refractory metastatic diseases showed that combined treatment of guadecitabine and cisplatin was tolerable and had demonstrated antitumor activity in platinum refractory GCTs [69]. A phase II non-randomized study analyzed 44 pediatric patients with refractory and recurrent non-testicular malignant GCTs to adopt simultaneous microwave-induced regional deep hyperthermia combined with PEI (cisplatin, etoposide and ifosphamide) chemotherapy and found that it might be useful for local disease control. However, the 5-year EFS and 5-year OS were 62% and 72%, respectively [70].

## 3. High Dose Chemotherapy and Autologous Stem Cell Transplantation in Pediatric GCTS

### 3.1. Overview

Most patients with stage I/II gonadal and stage III extragonadal diseases have a good treatment response to first-line treatment including surgery and chemotherapy. Patients with several clinical presentations have a poor prognosis, including adolescents, with extragonadal tumors, high tumor burden at diagnosis, and refractory to platinum-based chemotherapy [4,18]. The Malignant GCTs International Consortium (MaGIC) risk classification defined that females, aged more the 11 years, with stage IV extragonadal disease had an estimated long-term disease-free (LTDF) survival rate of approximately 40% [4]. Furthermore, poor-risk GCT subtypes such as CC often have metastatic disease at the time of diagnosis with a poor prognosis. Jiang et al. reported a median survival of 10 months after analyzing 113 adult male patients with extracranial CC [37]. The TGM 95 trial evaluated 19 patients with relapsed and refractory diseases; the 5 years EFS and OS were 26% and 32% respectively even after various chemotherapy and surgical treatments [57]. Even the radiation therapy or appropriate targeted therapy, based on the results of genetic alterations of the tumor were applied, but, in recent years, treatment results have not significantly improved the survival rate for patients with poor-risk or relapsed GCTs [64]. In order to improve the survival rate of patients with poor-risk GCTs, it might be beneficial to identify which patients are poor-risk and assess application of second-line salvage chemotherapy or adjuvant therapy. Table 1 lists poor-risk pediatric GCTs, defined as patients whose overall survival rate is less than 70%.

Since conventional standard dosage of chemotherapy has its limitations in the treatment of patients with poor-risk GCTs, HDCT combined with salvage ASCT has become the treatment option for adult patients with poor-risk or refractory GCTs. High dose carboplatin and etoposide (HD-CE) comprised the backbone of this treatment due to their comparable effectiveness and predominantly hematological toxicity which could be rescued with ASCT [71]. The effectiveness of HDCT and ASCT treatments for refractory or relapsed GCTs has also been demonstrated in recent reports [42,72,73,74]. 

We systematically analyzed HDCT combined with rescue ASCT to provide evidence-based studies for the treatment of patients with refractory and relapsed pediatric GCTs.

### 3.2. Autologous Stem Cell Transplantation Studies in Adult Patients

Metastatic GCTs are classified as favorable, intermediate and poor-risk groups according to the International Germ Cell Cancer Collaborative Group (IGCCCG). The proportion of the poor-risk group to achieve complete remission (CR) was less than 50% [75]. The chemo-sensitivity of GCTs is a strong rationale for testing high dose chemotherapy. However, this approach has been hampered by early death due to toxicity [76]. Most of the randomized trials failed to demonstrate significant benefit with HDCT and ASCT in high-risk frontline therapy patient groups [77]. However, Motzer et al. has shown that there was a subset of patients who had intermediate-to-high-risk GCTs with an unsatisfactory tumor marker decline after two cycles of BEP; these patients were switched to two cycles of HDCT and ASCT (carboplatin 600 mg/m^2^, etoposide 600 mg/m^2^, cyclophosphamide 50 mg/kg on day 1, day 2, and day 3, followed by infusion of autologous bone marrow on day 5). These patients may have a better outcome as compared with continued conventional treatment with BEP [78]. There was a two-year survival rate of 78% vs. 55% (*p* = 0.02), favoring the HDCT. However, toxicities were more severe in the BEP + HDCT arm [78]. The GETUG 13 supported how the slow decline of tumor markers in the poor-risk group are more likely to fail in conventional chemotherapy [41].

This emphasized that close monitoring of tumor markers when undergoing treatment may help to detect patients who will have poor responses earlier and show that a change in treatment strategy is needed. 

In adult patients with R/R diseases, Riaz et al. concluded that from two to three cycles of HDCT could improve survival [77]. For the difficult subtype of GCTs, Einhorn et al. analyzed 13 patients with CC and demonstrated that all patients who had progression disease after one to two lines of cisplatin combinations chemotherapy may respond to HDCT (carboplatin 700 mg/m^2^ and etoposide 750 mg/m^2^ for three consecutive days). Out of the 13 (46%) patients, 6 were alive and remained disease-free after 3 years of follow up [79]. In most studies, the conditioning regimen for malignant GCTs had no definitive guideline, but non-carboplatin- and etoposide (CE)-based HDCT failed to show benefits in relapse and refractory disease [77]. However, Gossi et al. also found that the poor response to treatment and mixed non-seminomatous tumors may be more likely to fail in the CE-based HDCT [80]. The idea to use paclitaxel when treating platinum-resistant cancer in general came from the experience of treating platinum-resistant ovarian cancer [81]. The Memorial Sloan–Kettering Cancer Center (MSKCC) study included 107 patients who used the TI-CE regimen (including two cycles of paclitaxel combined with ifosphamide for stem cell mobilization and followed by three cycles of high dose carboplatin and etoposide, which was a relatively low dose compared with the Indiana study) to overcome the toxicity of HDCT. The TI-CE regimens consist of paclitaxel and ifosphamide (TI) given 14 days apart and followed by 3 cycles of carboplatin and etoposide with autologous stem-cell support at intervals of between three and four weeks. Carboplatin dosing was with AUC 24 mg/mL/min, divided over three days, for the patients who received six and fewer cycles of cisplatin; patients who were more heavily pretreated received AUC 21 mg/mL/min divided over three days. Etoposide was given at a fixed dose of 400 mg/m^2^ from day 1 to day 3. The 5-year OS and 5-year DFS were 52% and 47%, respectively [82]. The same study also showed that the TI-CE regimen might overcome the platinum-resistant disease [82]. Adolescents with metastatic GCTs are biologically and clinically more similar to young adults than children; they are also more alike in outcomes [83,84]. The adolescents with metastatic GCTs may have benefited from this treatment modality.

### 3.3. Autologous Stem Cell Transplantation Studies in Pediatric Patients

Although most pediatric GCTs respond well to treatment, there are still some histopathologic subtypes, such as CC, that do not respond well to treatment and have a poor prognosis. For patients who do not respond well to treatment, even with multi-agent chemotherapy, including platinum followed by surgical resection, the long-term survival rate is only 10–20% [85]. To improve the survival of patients with poor-risk GCTs, the use of radiotherapy, target therapy, or HDCT with ASCT are treatment options based on the patient’s clinical and tumor molecular status. The following is a discussion of HDCT with ASCT in pediatric GCTs.

In the frontline therapy for children with solid tumors, such as high-risk neuroblastoma, increasing the intensity of chemotherapy drugs to improve the survival rate has been proven to be a feasible method [86]. However, previous pediatric GCTs reports have not been able to clearly define the role of HDCT as a frontline therapy. For example, 1 phase II trial involved 43 GCTs patients with chemotherapy cycles every 2 weeks instead of every 3 weeks. Although patients could tolerate the two-week interval cycle, the CR rate in the poor-risk group was still only 33%, and the 2-year PFS was 50% [87]. The above results suggest treatment results may be limited if chemotherapy is intensified by shortening the chemotherapy interval instead of increasing the dose of the regimen. The frontline HDCT therapy applied to pediatric GCTs may need to focus on poor-risk patients and is also worthy of further exploration.

There are several small observational studies on the treatment outcomes of children with R/R GCTs for HDCT. Giorgi et al. retrospectively studied 23 extragonadal GCTs in the European Blood and Marrow Transplantation (EBMT) database who received platinum-based salvage HDCT and found that 57% of R/R patients achieved CR, and the 5-year disease-free survival (DFS) rate was 43% without treatment-related death [74]. Another prospective French TGM95 study showed that 10 out of 19 R/R children with GCTs received carboplatin-based HDCT treatment, and four out of 10 (40%) patients treated with HDCT were alive as opposed to the two out of nine (22%) patients who were not treated with HDCT [42]. One recent cohort study included 18 children with R/R GCTs in Poland who were treated with melphalan-etoposide-carboplatin HDCT; they showed 5-year OS and 5-year EFS of 76% and 70.8%, respectively. Patients younger than 4 years of age who received HDCT combined with ASCT had an even higher survival rate [88]. Another recent AIEOP report considered 16 out of 21 children with R/R GCTs who received HDCT; 13 out of 16 (81%) patients were alive without treatment-related death [72]. The above reports confirm that most children with R/R GCTs respond to HDCT with ASCT and rarely have treatment-related mortality. All authors agree that HDCT following second-line conventional chemotherapy needs to be further investigated. Table 2 shows the reports of HDCT treatment in children with R/R GCTs.

One randomized phase III trial study, the TIGER trial (NCT 02375204), is ongoing. The primary aim of this trial is to compare the OS in patients treated with conventional-dose chemotherapy using the TIP regimen or the HDCT plus ASCT using the TI-CE regimen as initial salvation treatment of children with R/R GCTs. This study included children aged 14–18 [18,89]. We hope that this randomized control trial study will provide a better understanding of the application of HDCT in the treatment of children with R/R GCTs and the administration of conditioning regimens.

### 3.4. Conditioning Regimen

Four studies with retrospective and prospective data showed a great heterogeneity of conditioning regimen in current treatment of relapse and refractory malignant GCTs. (Table 2) From the study of the EBMT database, the conditioning regimen was quite varied. The conditioning regimen consisted of CarboPEC (carboplatin 250–350 mg/m^2^ for 4 days, etoposide 250–400 mg/m^2^ for 4 days and cyclophosphamide 1.6 g/m^2^ for 4 days), CE (carboplatin 250–500 mg/m^2^ for 3–4 days, etoposide 250–400 mg/m^2^ 3–4 days), and TE (thiotepa 300 mg/m^2^ for 3 days and etoposide 250–300 mg/m^2^ for 3 days). The 1-year OS and 1-year DFS was 74% and 52% [74]. For the first relapse patients, the 2-year OS was 78%. However, for the patients who had second-third relapse who underwent HDCT and ASCT, the 2-year OS was only 43% [74].

The conditioning regimen from the Italian group experience, using PEB (cisplatin 25 mg/m^2^ days 1 to 4, etoposide 100 mg/m^2^ day 1 to 4, bleomycin 15 mg/m^2^ day 2), was four courses as first line treatment. When R/R diseases occurred, ICE protocol (Ifosphamide: 1.8 g/m^2^ days 1 to 5, carboplatin 400 mg/m^2^ days 1 and 2, etoposide 100 mg/m^2^ days 1 to 5) as a second line treatment. The high-dose conditioning regimen was mostly based on thiotepa and cyclophosphamide. However, the details of the dose and duration of the chemotherapy were not mentioned. 

The most recent study in Poland analyzed 18 children prospectively. They included melphalan, etoposide, and carboplatin as a combination of HDCT regimens. The regimen consisted of melphalan 140 mg/m^2^ on day -6, etoposide 1800 mg/m^2^ on day -5, and carboplatin 500 mg/m^2^ on day -4, -3, and -2 (MEC 1). The original protocol was given to nine patients, and it was found that these patients had severe mucositis and life-threatening sepsis; the dose was therefore reduced to melphalan 140 mg/m^2^ on day-6, etoposide 200 mg/m^2^ on day-6 to -3, carboplatin 200 mg/m^2^ on days -6 to -3 (MEC 2). No toxic death occurred in this study. The 5-year OS and 5-year EFS was 76% and 70.8%. For the children younger than 4 years old, the OS and EFS were even higher [88].

## 4. Conclusions

Although most pediatric GCTs respond well to chemotherapy, the poor-risk group patients still have a worse outcome. Based on previously published reports on adults, or the few reports for children, treatment with HDCT combined with ASCT has a beneficial effect on R/R GCTs. However, the strategy of HDCT used as a frontline treatment has no demonstrated benefit for children in the poor-risk group and requires further research. At the same time, it is also necessary to consider the impact of HDCT related morbidity/mortality on prognosis. Therefore, it is important to identify those poor-risk patients who actually benefit from HDCT. For example, patients with tumor markers that do not decline or even increase during chemotherapy may represent a failure of front-line chemotherapy, and these patients may benefit from HDCT and ASCT treatment. Regarding the choice of conditioning regimens, HDCT based on CE (carboplatin and etoposide) is mostly used in adult R/R GCTs and is a treatment option for patients with platinum-refractory diseases. Since carboplatin and etoposide have been included in the front-line chemotherapy of pediatric GCTs, other drug combinations should be considered for conditioning regimens, such as melphalan, thiotepa, or paclitaxel. In conclusion, the treatment of pediatric poor-risk GCTs is still a challenge for pediatric oncologists. HDCT combined with ASCT in this group of patients requires further study.

## Figures and Tables

**Figure 1 cancers-15-01998-f001:**
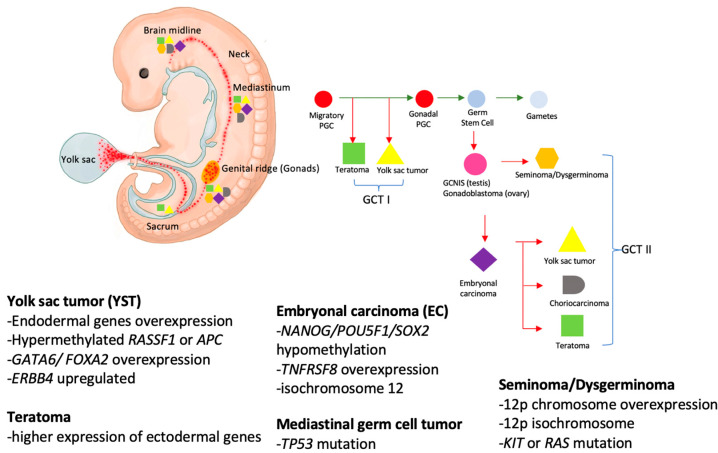
Primordial germ cell migration illustration and associated pathogenic gene mutations. Abbreviation: PGC: primordial germ cell; GCNIS: germ cell neoplasm in situ.

**Table 1 cancers-15-01998-t001:** Poor prognosis of germ cell tumor.

Disease Status	5-Year Overall Survival (%)
Ovarian stage IV disease and age ≥ 11 years	67 [4]
Extragonadal disease and age ≥ 11 years	
Stage II–III	65 [4]
Stage IV	40 [4]
Primary mediastinal germ cell tumors	54 [35]
Primary choriocarcinoma	- * [37]
Relapsed or refractory disease	32 [42]
Unfavorable tumor markers decline	- ** [41]

* 3-year OS 43.1%. ** 4-year OS 58%.

**Table 2 cancers-15-01998-t002:** Studies summarizing conditioning regimen for malignant germ cell tumors treatment.

Study	Country	Study Type	No. of Patients (n)	Median Age	HDC Regimen	Outcome (%)	Adverse Effects (%)
U De Giorgi et al. [74]	UK	Retrospective	23	12 years	CarboPEC	CR:16/23 (70)	TRM = 0 (0)
		EBMT database			CE	1y DFS: 52%	Grade 3 stomatitis = 9 (39)
					Thiotepa, VP-16	1y OS: 74%	Fever = 21 (81)
					CarboPETM	5y DFS:10/23 (43)	Infection = 13 (50)
							Grade 3 pulmonary toxicity = 2 (9)
							Grade 3 neurotoxicity = 1 (4)
							Psychosis = 1 (4)
							Veno-occlusion disease = 2 (9)
Cecile Faure-Conter et al. [42]	French	Prospective study	10	<20 years-old	CarboPEC* or	5y OS: 4/10 (40)	Not documented
					VP-16 and thiotepa		
De Pasquale et al. [72]	Italy	Retrospective study	16	21 months	VP-16, thiotepa, CY **	OS: 13/16 (81)	Grade 3 mucositis = 2 (13)
					Thiotepa, melphalan **		
Marek Ussowicz et al. [88]	Poland	Cohort study	18	<18 years-old	MEC1: 9	5y EFS: 70.8%	Leukopenia/neutropenia = 18 (100)
					MEC2: 9	5y OS: 76%	Fever = 15 (88)
							Grade 3 mucositis = 16 (88)
							Sepsis = 4 (22)
							Bacteremia = 1 (5)
							Veno-occlusion disease = 11 (61)

CarboPEC: Carboplatin 250–300 mg/m^2^ × 4 days or Calvert formula AUC = 7, Etoposide 250–400 mg/m^2^ × 4 days, cyclophosphamide 1.6 gm/m^2^ × 4 days. CE: carboplatin 250–500 mg/m^2^ × 3–4 days or Calvert formula AUC = 7, etoposide 250–400 mg/m^2^ × 3–4 days. TE: Thiothepa 300 mg/m^2^ × 3 days, Etoposide 250 mg-400 mg/m^2^ × 3–4 days. CarboPETM: Carboplatin 250–350 mg/m^2^ × 3–4 days or Calvert formula AUC = 7, etoposide 3–4 days, thiotepa 200–250 mg/m^2^ × 2–3 days, mephalan 80–100 mg/m^2^ × 1 day. CarboPEC*: carboplatin 250–550 mg/m^2^ on day 1, etoposide 450 mg/m^2^/day × 4 days, cyclophosphamide 1.6 gm/m^2^ × 4 days. VP-16 and thiotepa: Thiotepa 300 mg/m^2^ × 3 days, etoposide 500 mg/m^2^ × 3 days. MEC1: mephalan 140 mg/m^2^ on day -6, etoposide 1800 mg/m^2^ on day -5, carboplatin 500 mg/m^2^ on days × 3 days. MEC2: mephalan 140 mg/m^2^ on day -6, etoposide 200 mg/m^2^ on day × 4 days, carboplatin 200 mg/m^2^ on days × 4 days. TRM: treatment related mortality. ** not mentioned.

## Data Availability

The data can be shared up on request.

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
