# Peer review of "Pediatric Extracranial Germ Cell Tumors: Review of Clinics and Perspectives in Application of Autologous Stem Cell Transplantation"

_cancers, 2023, doi:10.3390/cancers15071998_

Round 1

Reviewer 1 Report

Germ cell tumors (GCTs) are rare in childhood, but a common malignancy in adolescents and young adults. The overall outcomes of patients treated for GCTs are excellent. The authors propose a review of the studies that evaluate the results of ASCT in refractory/relapsed patients with CGT and also the role in those at high risk. The current study presents very new and clearly presented data referring to pathogenesis, molecular biology and current classification of GCT. The therapeutic modalities for CGT are presented: chemotherapy, radiotherapy, targeted therapy. In the second part, the indications and results of ASCT in GCT in adults and children are presented, with a review of the most important data from the literature, and with the pertinent interpretation of each study. The article notes that ASCT is a well-established treatment option for patients with relapsed or refractory GCTs. Several studies have demonstrated high response rates and prolonged survival with this approach. Additionally, ASCT has been used in patients with high-risk disease, including those with poor prognostic factors such as elevated tumor markers or bulky disease. In these patients, ASCT has been shown to improve outcomes, including overall survival. The authors also discuss the role of ASCT as a consolidation therapy in GCTs. Several studies have evaluated the use of ASCT following standard chemotherapy in patients with high-risk disease. Overall, the review article concludes that ASCT is a valuable treatment option for patients with relapsed or refractory GCTs and those with high-risk disease. The authors note that further studies are needed to better define the optimal timing and patient selection for ASCT in this setting. In conclusion, the review article highlights the current place of ASCT in the treatment of GCTs. While ASCT has shown promise in various clinical settings, further studies are needed to define the optimal role of this approach in GCTs.

Author Response

Thank you for the review. We hope that our review the application of autologous stem cell transplantation in the treatment of pediatric germ cell tumors can provide the readers with more valuable treatment recommendations. 

Reviewer 2 Report

This is a well written and fairly comprehensive review of pediatric GCTs and H-dose therapy/ASCT.   There are been few papers and reviews in this area so it makes a good contribution to the literature.   There are a few typos that need to be fixed. 

Author Response

Thank you so much for the review. Please see the attachment. 

Reviewer 3 Report

1) General comments

This review article summarizes fundamental clinicopathological issues of extracranial germ cell tumors (GCTs) and possible clinical applications of high-dose chemotherapy combined with autologous stem cell transplantation for poor-risk tumors. The manuscript is well written, and includes comprehensive information from sufficient, unbiased references. I have only few comments for clarifying some aspects of the manuscript as below.

2) Specific comments

1. Germ cell neoplasia in situ (GCNIS) is a testis-specific lesion, which should be written in lines 91-91, lines 114-117, and Figure 1.

2. Lines 146-153: “Germinoma” is an intracranial GCT that is morphologically identical to its gonadal (testis, seminoma; ovary, dysgerminoma) and extragonadal (and extracranial) (seminoma) counterparts. This definition should be clearly described, and intracranial germinoma is not necessary to be focused on this extracranial GCT article.

3. References of the survival data are need to be added in Table 1.

4. Line 546: “EFS” or “DFS”? ;it is not consistent with the description in Table 2.

5. Line 549-555: reference should be added.

6. Table 2: “De Giogi” or “De Giorgi”?  “Pasquala” or “Pasquale”?  Please recheck author names and add reference numbers.
